# Biology-Informed Matrix Factorization: An AI-Driven Framework for Enhanced Drug Repositioning

**DOI:** 10.3390/biology14050549

**Published:** 2025-05-15

**Authors:** Yangyang Wang, Yaping Wang, Ya Hu, Jihan Wang

**Affiliations:** 1School of Physics and Electronic Information, Yan’an University, Yan’an 716000, China; 2Yan’an Medical College, Yan’an University, Yan’an 716000, China; 3Department of Medical College, Hunan Polytechnic of Environment and Biology, Hengyang 421000, China

**Keywords:** artificial intelligence, drug repositioning, non-negative matrix factorization, biological context, graph regularization, drug–disease association prediction

## Abstract

Finding new medical uses for existing drugs can help patients faster and at a lower cost compared to developing entirely new drugs. This process, called drug repositioning, uses the information we already know about approved drugs to see if they might treat other diseases. In this study, we created a computer model that combines drug and disease similarities based on biological knowledge, making predictions about which drugs might be effective for diseases they are not currently used to treat. Our method uses a mathematical technique called matrix factorization, enhanced by biological information, to improve accuracy and reliability. We tested the model using well-known data and showed that it works better than existing approaches. We also looked at several real drugs and found new disease connections for each one, many of which were supported by trusted medical databases. This approach helps researchers discover new treatment options faster and more efficiently, which could benefit public health and reduce costs in drug development.

## 1. Introduction

Drug repositioning, also known as drug repurposing, is the process of identifying new therapeutic uses for existing drugs beyond their original medical indications [1,2]. This strategy leverages established pharmacokinetic, pharmacodynamic, and toxicity profiles to accelerate development, reduce costs, and lower the risk of failure in clinical trials. Repositioning can involve both approved drugs and those discontinued for reasons unrelated to safety. The pharmaceutical industry continues to face substantial challenges in traditional drug discovery, including high costs, extended timelines, and high attrition rates [2]. Drug repositioning offers a practical alternative by reassessing existing compounds for new indications [3]. Compared to de novo drug development, which may take over a decade and cost upwards of $2 billion, repositioning significantly reduces both development time and financial burden. Moreover, as repositioned drugs have already undergone preclinical and clinical evaluations, they carry a lower risk of safety-related failure [4,5].

In recent years, a variety of computational approaches have been proposed to facilitate drug repositioning, including signature matching, network-based models, and matrix decomposition-based techniques [4]. Signature matching compares gene expression profiles of diseases and drugs to identify therapeutic candidates that may reverse disease phenotypes. Network-based methods explore biological interaction networks (e.g., protein–protein interaction, gene regulation) to uncover latent links between drugs and diseases. Matrix decomposition methods—such as non-negative matrix factorization (NMF) [6,7,8] and singular value decomposition (SVD) [9,10]—aim to extract low-rank representations from drug–disease association matrices, revealing latent associations that are often imperceptible in the high-dimensional space [11].

Among the above approaches, NMF has attracted significant attention due to its scalability, interpretability, and effectiveness in identifying hidden structures. For example, Zhang et al. proposed feature-derived graph regularized matrix factorization (FGRMF) [12], which predicts drug side effects by incorporating drug similarity features. The similarity-constrained matrix factorization method (SCMFDD) [13] extended this approach by adding regularization terms for both drug and disease similarity. Sadeghi et al. introduced NMFDR [6], a network-based NMF variant for predicting new drug–disease associations. Other recent developments include bounded nuclear norm regularization (BNNR) [14], which mitigates cold start problems, and hybrid models that combine matrix factorization with semantic diffusion frameworks [15]. However, existing NMF-based methods often neglect the biological context underlying drug–disease associations, focusing primarily on numerical decompositions without integrating domain-specific knowledge. This omission may limit both prediction accuracy and the interpretability of results.

To address these limitations, we propose a novel non-negative matrix factorization-based drug repositioning model that incorporates biological context (NMFIBC). Our framework embeds biological similarity networks—capturing functional or semantic relationships among drugs and diseases—into a graph-regularized NMF optimization scheme. This design enhances predictive accuracy and ensures that inferred associations are not only statistically robust but also biologically meaningful.

## 2. Materials and Methods

### 2.1. Datasets

We employed two gold-standard datasets (Cdataset [16] and Fdataset [17]) to evaluate the performance of the proposed method in predicting novel drug–disease associations. These datasets vary in size, containing different numbers of drugs, diseases, and known associations, as detailed in Table 1. Each dataset includes precomputed drug similarity matrices, disease similarity matrices, and a binary association matrix that encodes known drug–disease interactions.

### 2.2. Standard NMF

Suppose we have a set of n drugs and m disease, denoted as $D=\{d_{1},d_{2},...,d_{n}\}$ and $S=\{s_{1},s_{2},...,s_{m}\}$, respectively. The known side effects of these drugs, that is, the drug–side effect associations, are naturally represented by an n×m matrix of A. As illustrated in Figure 1A, the binary value “0” or “1” of entry $A_{i,j}$ indicates the absence or presence of disease $s_{j}$ for drug $d_{i}$. Matrix factorization plays a pivotal role in various applications, particularly in recommendation systems, data analysis, and machine learning. Its primary function is to decompose a large, complex matrix into a product of two or more smaller, lower-rank matrices, as shown in Figure 1B. The matrix of A can be decomposed into two low-rank matrices X and Y, where $A\approx X^{T}Y,X\in R^{k\times n},Y\in R^{k\times m}$ and rank(X)=rank(Y). Generally, the selection of the parameter may affect the accuracy of the final drug repositioning. From Figure 1, it is evident that the drug repurposing problem can be addressed through matrix factorization. This approach leverages the inherent structure of the data to decompose the complex relationships between drugs and their potential indications into more manageable components, thereby facilitating the discovery of new uses for existing drugs.

NMF is a valuable tool in data science due to its ability to provide insights into the composition of non-negative datasets and its interpretability of the underlying structure. The workflow of NMF is shown in Figure 1B. It is used to decompose a non-negative data matrix A into two or more non-negative matrices X and Y, whose product approximates the original matrix, and can be formulated by Equation (1):(1)$$\underset{W\geq 0,H\geq 0}{\min}||A-X^{T}Y|{|}_{F}^{2},s.t.X\geq 0,Y\geq 0$$
where $||\cdot |{|}_{F}$ is the Frobenius norm, $X\in R^{k\times n}$ and $Y\in R^{k\times m}$ (k≪min{m,n}) are the latent matrix, $X_{i}$ and $Y_{j}$ can be regarded as drug-specific and disease-specific latent feature vectors, and $A_{ij}=X_{i}^{T}Y_{j}$. There are several ways in which X and Y can be obtained, and the most popular method implemented by Lee et al. can be found in [18].

### 2.3. Proposed Model: NMFIBC

The standard NMF can sometimes be sensitive to noise, overfit the data, or fail to capture the underlying structure effectively. Tikhonov regularization can stabilize the NMF algorithm and ensure X,Y smoothness [19,20], especially when dealing with high-dimensional data or when the data are not well-represented by a low-rank approximation. The NMF with Tikhonov regularization model can be presented by using the following objective function:(2)$$\begin{array}{l} \min||A-X^{T}Y|{|}_{F}^{2}+\mu (||X|{|}_{F}^{2}+||Y|{|}_{F}^{2})\\ =\sum_{ij}{\left(A_{ij}-x_{i}^{T}y_{j}\right)}^{2}+\mu \left(\sum_{i}||x_{i}|{|}^{2}+{\sum_{j}\left||y_{j}|\right|}^{2}\right)\\ s.t.X\geq 0,Y\geq 0 \end{array}$$
where μ and λ are free parameters. By adjusting the regularization parameters, one can control the trade-off between fitting the data and maintaining the desired properties (like sparsity), allowing for more flexibility in model selection. Graph regularization can incorporate prior knowledge about the structure of the data, helping to preserve important relationships and patterns within the data [21,22]. In order to incorporate known drug interaction information, we developed an effective model, a non-negative matrix factorization-based drug repositioning method that incorporates biological context (NMFIBC). It amalgamates various data types: drug similarities $S_{d}$, disease similarities $S_{e}$, and established networks of drug–disease interactions to form a multifaceted network structure, as shown in Figure 2.

This model dissects matrix A into a pair of matrices with reduced rank, capturing the latent attributes of both drugs and diseases. Our model further imposes similarity constraints for drugs within these reduced-dimensional spaces. These drugs identified as potential candidates by the NMFIBC model may offer a foundation for deeper analytical investigation and subsequent empirical verification. Based on the drug similarity constraints and Equation (2), the objective function of NMFIBC can be formulated as Equation (3):(3)$$\begin{array}{l} L=\min||A-X^{T}Y|{|}_{F}^{2}+\mu (||X|{|}_{F}^{2}+||Y|{|}_{F}^{2})+\lambda (||X^{T}X-S_{d}|{|}_{F}^{2}+||Y^{T}Y-S_{e}|{|}_{F}^{2})\\ =\sum_{ij}{\left(A_{ij}-x_{i}^{T}y_{j}\right)}^{2}+\mu \left(\sum_{i}||x_{i}|{|}^{2}+{\sum_{j}\left||y_{j}|\right|}^{2}\right)\\ +\lambda (\sum_{i}\left||x_{i}^{T}x_{i}-S_{d}|{|}^{2}+\sum_{i}||y_{i}^{T}y_{i}-S_{e}|{|}^{2}\right)\\ s.t.X\geq 0,Y\geq 0 \end{array}$$

### 2.4. Optimization Algorithm

In this section, we use gradient descent to derive the solutions for the two latent feature matrices, X and Y, based on the objective function in Equation (3). In order to solve the optimization problem of Equation (3), we introduce Lagrange multipliers, $\Phi =[\varphi_{ik}]$ and $\Psi =[\psi_{jk}]$, to implement the constraints on X≥0,Y≥0 [23]. The objective function can be transformed into Equation (4).(4)$$\begin{array}{ll} L_{f} & =Tr(AA^{T}-AY^{T}X-X^{T}YA^{T}+X^{T}YY^{T}X)+\mu Tr(XX^{T}+YY^{T})\\ & +\lambda Tr(S_{d}S_{d}^{T}-S_{d}X^{T}X-X^{T}XS_{d}+X^{T}XX^{T}X)\\ & +\lambda Tr(S_{e}S_{e}^{T}-S_{e}Y^{T}Y-Y^{T}YS_{e}+Y^{T}YY^{T}Y)\\ & +Tr(\Phi X^{T})+Tr(\Phi Y^{T}) \end{array}$$

The partial derivatives with respect to X and Y are as Equations (5) and (6):(5)$$\frac{\partial L}{\partial X}=-YA^{T}+YY^{T}X+\mu X+2\lambda (-X(S_{d})+XX^{T}X)+\Phi$$(6)$$\frac{\partial L}{\partial Y}=-XA+XX^{T}Y+\mu Y+2\lambda (-Y(S_{e})+YY^{T}Y)+\Psi$$

By using the Karush–Kuhn–Tuker (KKT) conditions [24], Equations (5) and (6) can be transformed into Equations (7) and (8):(7)$${\left(-YA^{T}+YY^{T}X\right)}_{ik}X_{ik}+{\left(\mu X\right)}_{ik}X_{ik}+{\left(2\lambda (-XS_{d}+XX^{T}X)\right)}_{ik}X_{ik}=0$$(8)$${\left(-XA+XX^{T}Y\right)}_{jk}Y_{jk}+{\left(\mu Y\right)}_{jk}Y_{jk}+{\left(2\lambda (-YS_{e}+YY^{T}Y)\right)}_{ik}Y_{ik}=0=0$$

Thus, the updating rules for X and Y can be obtained, as shown in Formulas (9) and (10):(9)$$X_{ik}\leftarrow X_{ik}\frac{{\left(YA^{T}+2\lambda XS_{d}\right)}_{ik}}{{\left(YY^{T}X+\mu X+2\lambda XX^{T}X\right)}_{ik}}$$(10)$$Y_{jk}\leftarrow Y_{jk}\frac{{\left(XA+2\lambda YS_{e}\right)}_{jk}}{{\left(XX^{T}Y+\mu Y+2\lambda YY^{T}Y\right)}_{ik}}$$

Upon adequate convergence of X and Y, or upon reaching a predefined number of iterations, the predictive matrix $\overset{⌢}{A}$ can be accurately reconstructed as the product of obtained X and Y, $\overset{⌢}{A}\approx X^{T}Y$.

Algorithm 1 summarizes the procedure of NMFIBC for drug–disease association prediction.
**Algorithm 1:** Optimization algorithm for NMFIBC
**Input:** Drug similarity matrix, X∈Rn×n;Disease similarity matrix, Y∈Rm×m;
Drug-side effect association matrix, A∈Rn×m;The latent dimension of feature space, k<min⁡m,n;
Regularization parameter, μ>0, λ>0;Maximum iterations M.Output: The prediction matrix A^;Initialize X∈Rn×n and Y∈Rm×m;times ←0;**while** times<M **do**        update X by using Equation (9);        update Y by using Equation (10);         times←times+1;**end**
A^←XYT;

### 2.5. Evaluation Metrics

In this section, we present a comprehensive evaluation of the performance of five distinct algorithms (IMCMDA [25], NCPMDA [26], RLSMDA [27], and SIMCLDA [28]) on two datasets, Cdataset and Fdataset. The metrics of interest include Area Under the Curve (AUC), Area Under the Precision-Recall Curve (AUPR), Accuracy (Acc), Sensitivity or Recall (Sen), Specificity (Spe), Precision (Pre), and F1 Score (Fl), as shown in Equations (11)–(16). These metrics provide a holistic view of the classification capabilities of each algorithm. All the experiments were conducted using MATLAB 2023b on Windows 10, running on an Intel(R) Core (TM) i5-12400F at 2.50 GHz.(11)$$SN=\frac{TP}{TP+FN}$$(12)$$SP=\frac{TN}{TN+FP}$$(13)$$Acc=\frac{TP+TN}{TP+TN+FP+FN}$$(14)$$precision=\frac{TP}{TP+FP}$$(15)$$recall=\frac{TP}{TP+FN}$$(16)$$F1=\frac{2*precision*recall}{precision+recall}$$

## 3. Results

### 3.1. Performance Evaluation and Metric Analysis

To assess the predictive capability of the proposed NMFIBC model, we conducted a comparative analysis against four state-of-the-art algorithms (IMCMDA, NCPMDA, RLSMDA, and SIMCLDA) on two benchmark datasets: Cdataset and Fdataset. The evaluation metrics include AUC, accuracy, sensitivity, specificity, precision, and F1 score. As summarized in Table 2 and Table 3, NMFIBC achieves the highest AUC scores on both datasets (0.921 on Cdataset and 0.894 on Fdataset), outperforming all competing methods.

Beyond AUC, NMFIBC also demonstrates superior performance in accuracy (0.993 on Cdataset and 0.990 on Fdataset), along with consistently high precision and F1 scores. Notably, the model maintains a balanced trade-off between sensitivity and specificity, with specificity values exceeding 0.99, indicating robustness in distinguishing both positive and negative associations. In contrast, other models such as IMCMDA and RLSMDA show moderate accuracy but comparatively lower recall and F1 scores, suggesting potential overfitting or class imbalance issues. Figure 3 presents a visual comparison of AUC and AUPR scores across the five models, while Figure 4 illustrates their F1 scores. These results suggest that incorporating biological similarity networks through graph-regularized optimization enables NMFIBC to generalize more effectively, especially when dealing with sparse and noisy biomedical data.

### 3.2. Case Study Overview

To further assess the practical utility of NMFIBC, we conducted case studies on five representative drugs: Levodopa, Doxorubicin, Amantadine, Flecainide, and Tacrolimus. These drugs were selected due to their diverse clinical applications and well-documented pharmacological profiles. For each drug, we predicted the top five disease candidates that had no known associations in the original dataset. These predictions were ranked by their model-generated confidence scores and validated using biomedical databases such as DrugBank, Comparative Toxicogenomics Database (CTD), and Kyoto Encyclopedia of Genes and Genomes (KEGG). A summary of these case studies is provided in Table 4 (based on Cdataset) and Table 5 (based on Fdataset), which include the following: the original known associations (if any), the top five predicted diseases with scores, and supporting evidence from public databases or literature. These predictions serve as compelling demonstrations of how NMFIBC can uncover novel drug indications. Further interpretation and biological relevance of these results are provided in the Discussion Section.

## 4. Discussion

The increasing complexity of biological systems and the explosion of high-dimensional biomedical data have placed AI at the forefront of modern drug discovery and systems biology. In this context, the integration of AI algorithms with domain-specific biological knowledge has become essential for deriving meaningful insights from heterogeneous biomedical datasets. Our proposed model, a nonnegative matrix factorization-based drug repositioning framework that incorporates biological context (NMFIBC), is an example of such an approach. By embedding biological similarity networks into the matrix factorization process, NMFIBC addresses the limitations of traditional models and enhances the discovery of novel, biologically plausible drug–disease associations.

In these exploratory analyses, the NMFIBC algorithm was utilized to forecast novel therapeutic applications for existing medications in practical scenarios. During the discovery phase of fresh connections between pharmaceuticals and medical conditions, the existing links within a benchmark dataset served as our training material, while the uncharted pairs were designated as the pool of prospective associations. Upon employing the NMFIBC model to compute the predictive scores for the entire spectrum of potential drug–disease pairs, we ranked the candidate illnesses in a hierarchy based on the calculated scores specific to each medication. To ascertain the veracity of these forecasts, we handpicked Levodopa, Doxorubicin, Amantadine, Flecainide, and Tacrolimus as representative examples, scrutinizing the potential conditions forecasted by NMFIBC and subsequently cataloging the validation details for their top quintet of prospective illnesses. The associations of potential diseases with their respective drugs were corroborated through reputable public repositories, including DrugBank [42], CTD [43], and KEGG [44]. A synthesis of the prognostications and substantiating proof is encapsulated within Table 4 and Table 5.

Levodopa, a well-established treatment for Parkinson’s disease, is highlighted in both datasets. In the original Fdataset, the relationship between Levodopa and Parkinson’s disease (168600) was not explicitly documented. However, our model was able to accurately predict this known drug–disease link, demonstrating its robust predictive power. The Fdataset also predicts additional candidate diseases such as insensitivity to pain with hyperplastic myelinopathy (147530) and restless leg syndrome (102300), with weights indicating the strength of these predictions. The Cdataset reinforces Levodopa’s association with Parkinson’s disease and introduces Alzheimer’s disease (605055) as a candidate for further investigation. Doxorubicin, primarily used in cancer treatment, shows a robust existing link with diseases like small cell cancer of the lung (182280) and breast cancer (114480) in the Fdataset. The predictions extend to other cancer types, indicating a potential broad-spectrum activity against various malignancies. The Cdataset also emphasizes Doxorubicin’s connection with cancer, predicting lymphoblastic leukemia (247640) and testicular germ cell tumor (273300) as new candidates. Amantadine, traditionally used for influenza and Parkinson’s disease, is presented with a significant existing relationship with multiple sclerosis (126200) in the Cdataset. The Fdataset predicts restless legs syndrome (102300) and Alzheimer’s disease (104300) as new candidates, suggesting a potential neuroprotective role for Amantadine beyond its current uses. Flecainide, an antiarrhythmic medication, is shown in the Cdataset with a strong existing relationship with atrial fibrillation (607554) and is predicted to have potential against dermatitis (603165) and allergic rhinitis (607154) in the Fdataset. This suggests that Flecainide may have immunomodulatory properties that could be harnessed for non-cardiac conditions. Tacrolimus, an immunosuppressive drug, is associated with asthma (208550) and dermatitis (603165) in the Fdataset, hinting at its potential in treating autoimmune and inflammatory conditions. The Cdataset also points to Tacrolimus’s potential in treating dermatitis (605805) and asthma (600807), further supporting its broad application in immunological disorders.

These case studies serve not only as illustrative examples of the model’s predictive capabilities but also as practical validations that demonstrate its biological relevance in real-world settings. While the core strength of NMFIBC lies in its systematic integration of biological similarity networks to enhance drug–disease association prediction, the case-level findings provide complementary insight into how these computational predictions translate into meaningful hypotheses for drug repurposing. Collectively, these findings underscore the potential of intelligent, context-aware modeling frameworks to advance our understanding of pharmacological mechanisms and to support hypothesis-driven biomedical research. The integration of rigorous benchmarking with interpretable, biologically grounded predictions makes this approach especially valuable in navigating the complexity of modern therapeutic discovery.

## 5. Conclusions

These case studies serve not only as illustrative examples of the model’s predictive capabilities but also as practical validations that demonstrate its biological relevance in real-world settings. While the core strength of NMFIBC lies in its systematic integration of biological similarity networks to enhance drug–disease association prediction, the case-level findings provide complementary insight into how these computational predictions can generate meaningful and testable hypotheses for drug repurposing.

Collectively, these findings underscore the potential of intelligent, context-aware modeling frameworks to advance our understanding of pharmacological mechanisms and support hypothesis-driven biomedical research. By combining rigorous performance benchmarking with biologically grounded interpretability, the NMFIBC framework contributes to bridging the gap between computational predictions and translational applications in therapeutic discovery.

## Figures and Tables

**Figure 1 biology-14-00549-f001:**
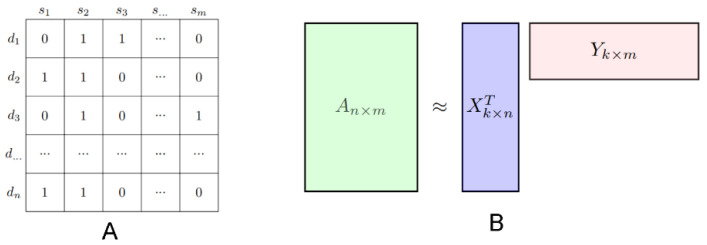
Overview of the standard non-negative matrix factorization (NMF) workflow for drug–disease association prediction. (**A**) Binary drug–disease association matrix representing known interactions. (**B**) Decomposition of the matrix into two lower-dimensional matrices capturing latent drug and disease features for use in predictive modeling.

**Figure 2 biology-14-00549-f002:**
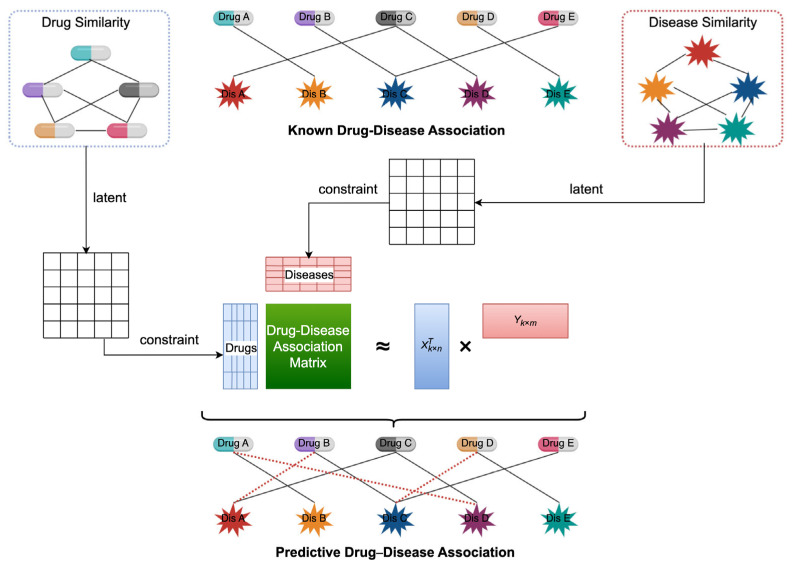
Workflow of the proposed NMFIBC framework integrating biological similarity networks. The model incorporates drug and disease similarity information into the matrix factorization process via graph-regularized constraints, enhancing prediction accuracy and biological relevance.

**Figure 3 biology-14-00549-f003:**
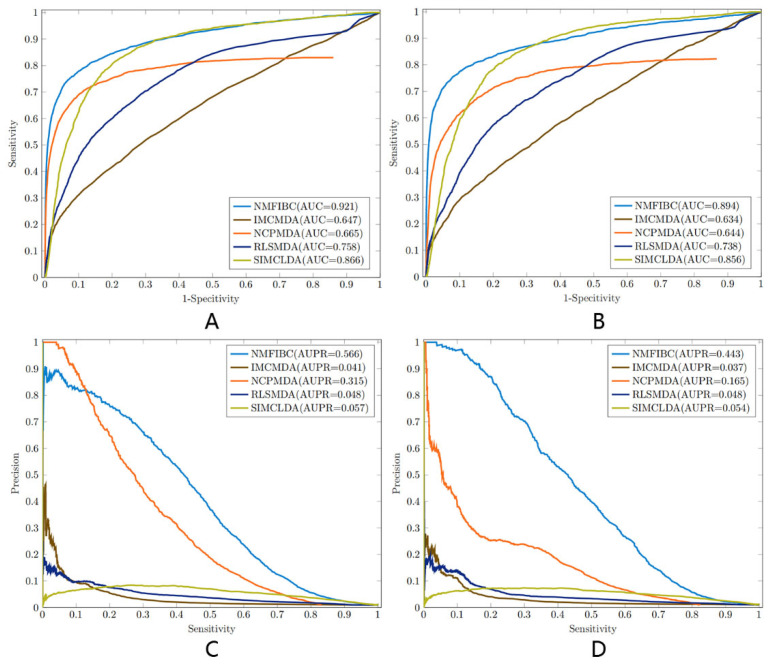
AUC and AUPR scores of the five models (NMFIBC, IMCMDA, NCPMDA, RLSMDA, and SIMCLDA) evaluated on Cdataset and Fdataset. (**A**,**B**) AUC for Cdataset and Fdataset, respectively. (**C**,**D**) AUPR for Cdataset and Fdataset, respectively.

**Figure 4 biology-14-00549-f004:**
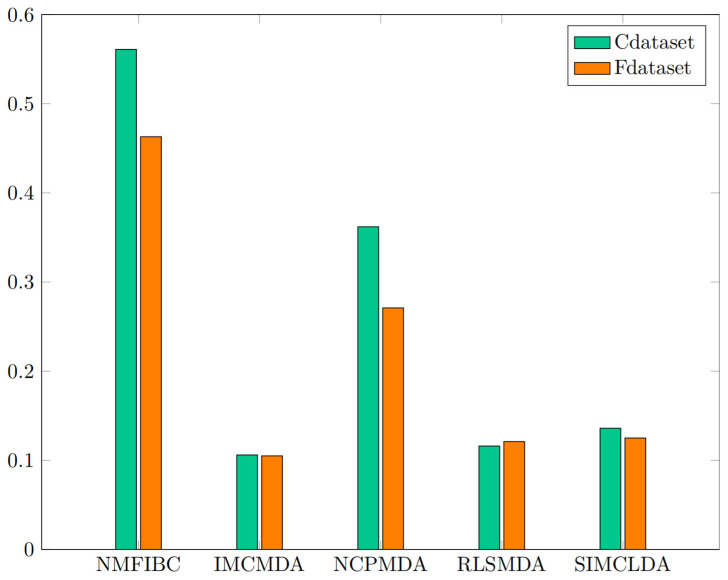
F1 Score of the five models on Cdataset and Fdataset.

**Table 1 biology-14-00549-t001:** Summary of drug–disease similarity matrices and known associations in the benchmark datasets.

Dataset	Drug Similarity Matrix	Disease Similarity Matrix	Known Associations
Cdataset	663 × 663	409 × 409	409 × 663
Fdataset	593 × 593	313 × 313	313 × 593

**Table 2 biology-14-00549-t002:** Performance comparison of competing methods on Cdataset across multiple evaluation metrics.

	AUC	AUPR	Acc	Sen (Recall)	Spe	Pre	F1
NMFIBC	0.921	0.566	0.993	0.504	0.997	0.633	0.561
IMCMDA	0.647	0.041	0.979	0.129	0.988	0.090	0.106
NCPMDA	0.665	0.315	0.989	0.333	0.995	0.395	0.362
RLSMDA	0.758	0.048	0.979	0.149	0.987	0.096	0.116
SIMCLDA	0.866	0.057	0.954	0.392	0.959	0.083	0.136

**Table 3 biology-14-00549-t003:** Performance comparison of competing methods on Fdataset across multiple evaluation metrics.

	AUC	AUPR	Acc	Sen (Recall)	Spe	Pre	F1
NMFIBC	0.894	0.443	0.990	0.430	0.996	0.501	0.463
IMCMDA	0.634	0.037	0.982	0.100	0.992	0.111	0.105
NCPMDA	0.644	0.165	0.981	0.343	0.988	0.224	0.271
RLSMDA	0.738	0.048	0.982	0.121	0.991	0.121	0.121
SIMCLDA	0.856	0.054	0.941	0.408	0.946	0.074	0.125

**Table 4 biology-14-00549-t004:** Top five predicted diseases for selected drugs based on Cdataset and their validation using public databases.

Drugs	Diseases (Existing Relations in Original Matrix)	Top Five Predicted Candidate Diseases (No Relation in Original Matrix)	Weight	Evidence
Levodopa (DB01235)	Paralysis agitans (168100) Parkinson disease (168600) Parkinson disease 2 (600116) Parkinson disease 7 (606324) Parkinson disease 15 (260300)	Dementia (125320)	0.761	DB/KEGG
		Alzheimer disease 9 (608907)	0.571	DB/KEGG
		Alzheimer disease (605055)	0.568	DB/KEGG
		Alzheimer disease 2 (104310)	0.560	DB/KEGG
		Alzheimer disease 5 (602096)	0.536	DB/KEGG
Doxorubicin (DB00997)	Mismatch repair cancer syndrome 1 (276300) Breast cancer (114480) Lymphoblastic leukemia (247640) Leukemia (601626) Lymphoma (236000)	Renal cell carcinoma (144700)	0.734	DB/KEGG
		Testicular germ cell tumor (273300)	0.692	DB
		Small cell cancer of the lung (182280)	0.654	DB
		Leukemia (246470)	0.651	KEGG
		Dohle bodies and leukemia (223350)	0.649	KEGG
Amantadine (DB00915)	Paralysis agitans (168100) Multiple sclerosis (126200) Popliteal pterygium syndrome (119500)	Parkinson’s disease 7 (606324)	0.337	DB/KEGG/CTD
		Parkinson’s disease 15 (260300)	0.325	DB/KEGG/CTD
		Schizophrenia (181500)	0.322	DB/KEGG
		Parkinson’s disease (168600)	0.318	DB/KEGG/CTD
		Parkinson’s disease 2 (600116)	0.318	DB/KEGG/CTD
Flecainide (DB01195)	Atrial fibrillation (607554)	Hypertension (608622)	0.688	[29]
		Renal failure (161900)	0.672	[29]
		Insensitivity to pain with hyperplastic Myelinopathy (147530)	0.520	Unknown
		Raynaud disease (179600)	0.413	Unknown
		Atrial fibrillation (608583)	0.404	DB/KEGG/CTD
Tacrolimus (DB00864)	Dermatitis (603165) Dermatitis (605805) Dermatitis (605804) Dermatitis (605844)	Allergic rhinitis (607154)	0.625	[30]
		Asthma (208550)	0.462	[31]
		Asthma (600807)	0.438	[31]
		Breast cancer (114480)	0.424	[32]
		Renal failure (161900)	0.396	[33]

**Table 5 biology-14-00549-t005:** Top five predicted diseases for selected drugs based on Fdataset and their validation using public databases.

Drugs	Diseases (Existing Relations in Original Matrix)	Top Five Predicted Candidate Diseases (No Relation in Original Matrix)	Weight	Evidence
Levodopa (DB01235)	Paralysis agitans (168100) Restless legs syndrome (102300)	Parkinson’s disease (168600)	0.548	DB/KEGG/CTD
		Insensitivity to pain with hyperplastic Myelinopathy (147530)	0.531	Unknown
		Dementia (125320)	0.451	DB/KEGG, [34]
		Renal failure (161900)	0.422	Unknown
		Attention deficit hyperactivity disorder (143465)	0.382	Unknown
Doxorubicin (DB00997)	Myeloma (254500) Breast cancer (114480) Neuroblastoma (256700) Leukemia (601626) Lymphoma (236000)	Small cell cancer of the lung (182280)	0.577	[35]
		Colorectal cancer (114500)	0.573	[36]
		Testicular germ cell tumor (273300)	0.530	[37]
		Kaposi sarcoma (148000)	0.518	DB/KEGG
		Esophageal cancer (133239)	0.513	[38]
Amantadine (DB00915)	Paralysis agitans (168100) Multiple sclerosis (126200) Popliteal pterygium syndrome (119500)	Dementia (125320)	0.365	DB/KEGG/CTD
		Parkinson’s disease (168600)	0.363	DB/KEGG/CTD
		Restless legs syndrome (102300)	0.295	[39]
		Alzheimer’s disease (104300)	0.227	DB/KEGG/CTD
		Alzheimer disease (605055)	0.216	DB/KEGG/CTD
Flecainide (DB01195)	Atrial fibrillation (607554)	Hypertension (608622)	0.597	[29]
		Renal failure (161900)	0.560	[29]
		Atrial fibrillation (608583)	0.524	DB/CTD, [29]
		Insensitivity to pain with hyperplastic Myelinopathy (147530)	0.463	Unknown
		Stroke (601367)	0.335	[40]
Tacrolimus (DB00864)	Dermatitis (603165)	Renal failure (161900)	0.582	[33]
		Hypertension (608622)	0.490	[41]
		Asthma (208550)	0.381	[31]
		Insensitivity to pain with hyperplastic Myelinopathy (147530)	0.376	Unknown
		Hypoparathyroidism (146255)	0.374	Unknown

## Data Availability

The raw data supporting the conclusions of this article will be made available by the authors upon request.

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
