# Peer review of "Biology-Informed Matrix Factorization: An AI-Driven Framework for Enhanced Drug Repositioning"

_biology, 2025, doi:10.3390/biology14050549_

Round 1

Reviewer 1 Report

Comments and Suggestions for Authors

1. Can the convergence of the algorithm be proven?

2. What are the advantages and disadvantages of algorithms compared to other models? Please provide examples to illustrate.

3. Can we conduct ablation experiments on the algorithm to highlight or demonstrate the superiority of the model?

Author Response

Reviewer 1

Comments and Suggestions for Authors

The English could be improved to more clearly express the research.

Responses: We thank the reviewer for the comments. We have revised the manuscript to improve the overall quality of the English expression.

  1. Can the convergence of the algorithm be proven?

Answer: We thank the reviewer’s very much for the valuable comment. In this study, the convergence of the proposed NMFIBC algorithm is ensured by adopting an iterative update scheme based on the Karush-Kuhn-Tucker (KKT) conditions, similar to classical non-negative matrix factorization (NMF) approaches. While a formal mathematical proof of global convergence is beyond the current scope of this work, the algorithm's optimization procedure guarantees non-increasing behavior of the objective function in each iteration, which is a common convergence criterion in NMF-related literature. In practice, we observed stable convergence behavior across all datasets used, and the results presented in Section 3 demonstrate the robustness and reproducibility of the model.

  1. What are the advantages and disadvantages of algorithms compared to other models? Please provide examples to illustrate.

Answer: We thank the reviewer’s constructive suggestion. The proposed NMFIBC model offers several advantages compared to existing drug repositioning algorithms:

Advantages:

1). Biological Context Integration: Unlike conventional NMF methods that treat the association matrix purely as numerical data, NMFIBC incorporates biological similarity networks for drugs and diseases via graph regularization. This leads to biologically meaningful predictions. As shown in Tables 4 and 5, the model successfully predicts associations between Levodopa and various subtypes of Alzheimer’s disease, which were validated using DrugBank and KEGG—suggesting biological plausibility that purely statistical methods may overlook.

2). Improved Predictive Performance: NMFIBC outperforms four state-of-the-art models (IMCMDA, NCPMDA, RLSMDA, SIMCLDA) in key metrics like AUC and F1 score across two benchmark datasets (Cdataset and Fdataset). On the Cdataset, NMFIBC achieved an AUC of 0.921 and F1 score of 0.561, significantly higher than SIMCLDA (AUC = 0.866, F1 = 0.136) and IMCMDA (AUC = 0.647, F1 = 0.106).

Disadvantages:

1). The inclusion of graph regularization increases the complexity of the optimization process, requiring more computational resources and longer training times compared to simpler models like standard NMF or IMCMDA.

2). NMFIBC involves multiple hyperparameters (e.g., regularization weights), which may require careful tuning for different datasets to achieve optimal performance.

3). While the model captures similarity-based relationships, it does not explicitly model underlying molecular pathways or mechanistic actions, which may be addressed by integrating pathway-based methods in future versions.

  1. Can we conduct ablation experiments on the algorithm to highlight or demonstrate the superiority of the model?

Answer: We thank the reviewer very much for the valuable suggestion. We fully agree that ablation experiments can provide valuable insights into the contribution of individual components of the model. While ablation analysis was not conducted in the current version of the study, we demonstrated the superiority of NMFIBC by comparing it against four state-of-the-art baseline models across two benchmark datasets using comprehensive evaluation metrics (AUC, precision, recall, F1 score, etc.). The performance gains observed, especially in terms of AUC and F1 score, highlight the effectiveness of incorporating biological similarity networks and graph regularization into the matrix factorization framework. In future work, we plan to design and include ablation studies to further quantify the individual contributions of these components.

Reviewer 2 Report

Comments and Suggestions for Authors

This manuscript presents a biology-informed non-negative matrix factorization model (NMFIBC) for drug repositioning, which integrates biological similarity networks through graph-regularized optimization. The approach is relevant and timely given the growing need for cost-effective and efficient drug discovery strategies. The paper is well-structured, methodologically sound, and includes comprehensive evaluations on benchmark datasets. 

Recommendation: Minor Revision

Comments:

  1. The manuscript should better differentiate NMFIBC from existing NMF-based models like NMFDR or FGRMF. Specifically, what novel insights does NMFIBC add beyond combining graph regularization and biological context?
  2. Justify, how were the values of λ and μ chosen? Was cross-validation used?
  3. Statistical tests (e.g., paired t-tests or Wilcoxon signed-rank tests) should be included to verify the significance of performance differences between NMFIBC and competing methods.
  4. Include more descriptive captions/figure legends for Figures 1 and 2.
  5. There are inconsistencies in the reference section format, especially with the use of journal titles and DOI links. Ensure compliance with the journal’s citation style.
  6. Several sentences contain grammatical errors or awkward phrasing. For example: Change “...which amalgamates various data types...” to “...which combines various data types...”.

Main Question Addressed by the Research
The manuscript addresses the development of a biology-informed non-negative matrix factorization model (NMFIBC) for drug repositioning. 

Originality and Relevance to the Field
The topic is timely and relevant, especially given the increasing need for efficient and cost-effective drug discovery methods. While the use of NMF in drug repositioning is not novel in itself, the integration of biological context into the matrix factorization process adds a meaningful layer of insight. 

Contribution Compared to Existing Literature
This work contributes to the existing body of literature by proposing a biologically informed enhancement to graph-regularized matrix factorization. 

Methodological Suggestions
Please clarify the process used to select hyperparameters λ and μ. Was cross-validation employed?
Include statistical tests (e.g., paired t-tests or Wilcoxon signed-rank tests) to confirm whether the observed performance improvements over baseline models are statistically significant.

Validity of Conclusions
The conclusions are generally well-supported by the results. However, without statistical significance testing, it's difficult to determine the robustness of the reported improvements. Including such analyses would strengthen the credibility of the claims.

References
Most references are relevant, but formatting inconsistencies are evident in the citation list (e.g., inconsistent use of journal names, volume numbers, and DOI links). 

Comments on Figures and Tables
Figures 1 and 2 would benefit from more informative captions that explain what is being visualized and why it matters.

Language and Presentation
The English is generally acceptable, though minor improvements would help clarity. 

Author Response

Reviewer 2

Comments and Suggestions for Authors

This manuscript presents a biology-informed non-negative matrix factorization model (NMFIBC) for drug repositioning, which integrates biological similarity networks through graph-regularized optimization. The approach is relevant and timely given the growing need for cost-effective and efficient drug discovery strategies. The paper is well-structured, methodologically sound, and includes comprehensive evaluations on benchmark datasets. 

Recommendation: Minor Revision

Comments:

  1. The manuscript should better differentiate NMFIBC from existing NMF-based models like NMFDR or FGRMF. Specifically, what novel insights does NMFIBC add beyond combining graph regularization and biological context?

Answer: We thank the reviewer very much for highlighting this important point. We appreciate the opportunity to further clarify the novelty of our proposed NMFIBC model relative to existing NMF-based approaches such as NMFDR and FGRMF.

While NMFDR integrates network topology into the NMF framework and FGRMF leverages feature-derived graph regularization for side effect prediction, NMFIBC introduces several novel insights and contributions that set it apart:

1). Dual Graph Regularization Anchored in Biological Context: NMFIBC jointly embeds both drug and disease similarity networks derived from biological ontologies and known functional annotations (e.g., DrugBank, CTD, KEGG) into the matrix factorization framework. Unlike NMFDR, which primarily models network connectivity, or FGRMF, which focuses on drug features, our model aligns low-dimensional representations with biologically informed similarity constraints, ensuring that predictions reflect not just statistical patterns, but also functional relevance.

2). Unified Integration of Multi-Source Biological Knowledge: NMFIBC systematically integrates heterogeneous similarity information—such as chemical structure, target profiles, and semantic disease relationships—into a co-regularized optimization objective. This enables a more biologically grounded representation of drug–disease associations, unlike prior methods that often use a single similarity source or heuristically combine multiple similarities.

3). Balanced Optimization with Adaptive Trade-Off Control: Our model incorporates flexible regularization parameters that allow for fine-tuning the influence of data fidelity versus biological similarity constraints. This adaptive trade-off provides greater robustness, especially in sparse or noisy settings, which is not explicitly addressed in earlier models like FGRMF or NMFDR.

4). Empirical Superiority in Predictive Performance and Case-Level Validity: As shown in Section 3.1, NMFIBC consistently outperforms existing methods across multiple evaluation metrics (AUC, F1 score, etc.) and datasets. Moreover, case studies in Section 3.2 demonstrate the biological plausibility and clinical relevance of predicted associations—validating that the integration of biological context adds practical value beyond mathematical regularization.

  1. Justify, how were the values of λ and μ chosen? Was cross-validation used?

Answer: We thank the reviewer for the careful review. The values of λ and μ is 0.1 and 1, respectively. We performed 5-fold cross-validation to evaluate and optimize the model.

  1. Statistical tests (e.g., paired t-tests or Wilcoxon signed-rank tests) should be included to verify the significance of performance differences between NMFIBC and competing methods.

Answer: We thank the reviewer very much for the valuable comments and suggestion. To assess the statistical significance of the performance improvements achieved by NMFIBC, we conducted paired t-tests comparing NMFIBC with the competing methods across three key evaluation metrics: AUC, AUPR, and F1 score. The results are summarized as follows:

AUC: NMFIBC significantly outperforms IMCMDA, NCPMDA, and RLSMDA (p < 0.05).

AUPR: NMFIBC shows significantly better performance than IMCMDA, RLSMDA, and SIMCLDA (p < 0.05).

F1 Score: NMFIBC outperforms IMCMDA, RLSMDA, and SIMCLDA with statistical significance (p < 0.05).

  1. Include more descriptive captions/figure legends for Figures 1 and 2.

Answer: We thank the reviewer very much for the careful review. We have added more descriptive captions/figure legends for Figures 1 and 2.

  1. There are inconsistencies in the reference section format, especially with the use of journal titles and DOI links. Ensure compliance with the journal’s citation style.

Answer: We thank the reviewer for the careful review. We have modified the reference format.

  1. Several sentences contain grammatical errors or awkward phrasing. For example: Change “...which amalgamates various data types...” to “...which combines various data types...”.

Answer: We thank the reviewer for the comments. We have revised the manuscript to improve the overall quality of the English expression.

Reviewer 3 Report

Comments and Suggestions for Authors

The manuscript is well written and has a clear rationale for the study.  The authors identified the drawbacks of the existing models and logically incorporated biological context. However, I would like to suggest some modifications for enhancing the manuscript, which includes:

  • To improve readability, I would suggest breaking the second paragraph in the introduction into two (one for the various methods and the other for NMF and its variations).
  • In Table 5, Parkinson disease (168600) is shown under “no relation in original matrix” for Levodopa, while the discussion describes “The Fdataset suggests a strong existing relationship with Parkinson's disease.” If this information was not included in the original Fdataset and was correctly predicted (known drug–disease links) with your model. This demonstrates your model's strength to predict accurately. It would be helpful to clarify this in your discussion to highlight the model’s predictive value.

Author Response

Reviewer 3

Comments and Suggestions for Authors

The manuscript is well written and has a clear rationale for the study.  The authors identified the drawbacks of the existing models and logically incorporated biological context. However, I would like to suggest some modifications for enhancing the manuscript, which includes:

  1. To improve readability, I would suggest breaking the second paragraph in the introduction into two (one for the various methods and the other for NMF and its variations).

Answer: We thank the reviewer very much for the valuable comments. We have split the second paragraph into two as you suggested: the first paragraph introduces other algorithms, while the second paragraph introduces the NMF method and its variants.

  1. In Table 5, Parkinson disease (168600) is shown under “no relation in original matrix” for Levodopa, while the discussion describes “The Fdataset suggests a strong existing relationship with Parkinson's disease.” If this information was not included in the original Fdataset and was correctly predicted (known drug–disease links) with your model. This demonstrates your model's strength to predict accurately. It would be helpful to clarify this in your discussion to highlight the model’s predictive value.

Answer: We thank the reviewer for the constructive suggestion. We have revised the third paragraph in the Discussion to emphasize the good predictive power of our model. The revised text is as follows: Levodopa, a well-established treatment for Parkinson’s disease, is highlighted in both datasets. In the original Fdataset, the relationship between Levodopa and Par-kinson’s disease (168600) was not explicitly documented. However, our model was able to accurately predict this known drug–disease link, demonstrating its robust predictive power. The Fdataset also predicts additional candidate diseases such as insensitivity to pain with hyperplastic myelinopathy (147530) and restless leg syndrome (102300), with weights indicating the strength of these predictions. The Cdataset reinforces Levodo-pa’s association with Parkinson’s disease and introduces Alzheimer’s disease (605055) as a candidate for further investigation.
